# Deep learning model for personalized prediction of positive MRSA culture using time-series electronic health records

Masayuki Nigo ®[1,2,3] ✉, Laila Rasmy ®[2], Bingyu Mao ®[2], Bijun Sai Kannadath ®[4], Ziqian Xie[2] & Degui Zhi ®[2]

Methicillin-resistant *Staphylococcus aureus* (MRSA) poses significant morbidity and mortality in hospitals. Rapid, accurate risk stratification of MRSA is crucial for optimizing antibiotic therapy. Our study introduced a deep learning model, PyTorch_EHR, which leverages electronic health record (EHR) time-series data, including wide-variety patient specific data, to predict MRSA culture positivity within two weeks. 8,164 MRSA and 22,393 non-MRSA patient events from Memorial Hermann Hospital System, Houston, Texas are used for model development. PyTorch_EHR outperforms logistic regression (LR) and light gradient boost machine (LGBM) models in accuracy ($AUROC^{PyTorch\_EHR} = 0.911$, $AUROC^{LR} = 0.857$, $AUROC^{LGBM} = 0.892$). External validation with 393,713 patient events from the Medical Information Mart for Intensive Care (MIMIC)-IV dataset in Boston confirms its superior accuracy ($AUROC^{PyTorch\_EHR} = 0.859$, $AUROC^{LR} = 0.816$, $AUROC^{LGBM} = 0.838$). Our model effectively stratifies patients into high-, medium-, and low-risk categories, potentially optimizing antimicrobial therapy and reducing unnecessary MRSA-specific antimicrobials. This highlights the advantage of deep learning models in predicting MRSA positive cultures, surpassing traditional machine learning models and supporting clinicians' judgments.

Methicillin-resistant *Staphylococcus aureus* (MRSA) is a common pathogenic cause of hospital-acquired and community-associated infections[1–3]. Since this pathogen eliminates most beta-lactam class antibiotics as a treatment option, physicians often need to add an antibiotic, such as vancomycin, to empirically treat this pathogen when suspected. Considering the side effect profile of vancomycin and the antibiotic stewardship standpoint, avoiding unnecessary antimicrobial therapy is highly desirable[4]. Furthermore, a recent study showed the absolute benefit of empiric therapy against MRSA is 0.1% or less[5]. Therefore, accurately identifying high-risk patients is critical to preserve the benefit of treatment and minimize the adverse side effects of empiric therapy. Although multiple clinical factors have been

proposed as risk factors for MRSA infection[6–9], there are several limitations to identifying high-risk patients. Commonly, the tested population is restricted to specific populations, such as patients with ventilator-associated pneumonia[10]. Due to the complex associations among risk factors, it is often difficult to discern actual risks when multiple risk factors exist simultaneously[8]. For example, previous exposure to cephalosporine and fluoroquinolone are considered risk factors[11,12]. The risk seems to accumulate when multiple antibiotics are previously prescribed[13]. Furthermore, the optimal timeline between the index infection and the presence of the risk factor is not well established, and often an arbitrary duration is used[8,14]. More flexible models that can integrate multiple risk factors and the timing of

[1]McGovern Medical School, University of Texas Health Science Center at Houston, Houston, TX, USA. [2]McWilliams School of Biomedical Informatics, University of Texas Health Science Center at Houston, Houston, TX, USA. [3]Division of Infectious Diseases, Department of Medicine, Houston Methodist Hospital, Texas Medical Center, Houston, TX, USA. [4]Department of Internal Medicine, University of Arizona College of Medicine, Phoenix, AZ, USA. ✉ e-mail: mnigo@houstonmethodist.org

various risk factors are warranted for frontline physicians to safely decide the necessity of empiric antibiotic therapy.

Electronic health records (EHRs) became widely available in the United States since the Meaningful Use program was introduced as part of the 2009 Health Information Technology for Economic and Clinical Health Act[15]. EHRs are a rich data source for daily clinical practice and research. As the data in EHRs expand, physicians have more information to process and interpret to improve patient management. Given its computational capabilities, artificial intelligence could reveal complex relationships among numerous factors in EHRs. Artificial intelligence has been used to process genetic and imaging data and has become an attractive technology to process real-time big EHR data to facilitate personalized medicine[16,17]. Although there are multiple machine learning models predict drug-resistant bacterial infections with EHR data[18–20], they use limited input data, such as basic demographics, previous susceptibility results, or a limited number of patients[18]. Furthermore, some models only predict the index culture or screening results, which may not be optimal in clinical use to guide antibiotic therapy[21]. Deep learning-based models, such as recurrent neural network (RNN) models, have a significant advantage in time-sequence events because the fundamental model structure allows sequential inputs into the model. Also, RNNs with medical code embedding can take inputs directly from a real-time EHR data stream, automatically adjust to reflect subtle changes, and provide real-time outputs[22] PyTorch_EHR, a deep learning-based prediction model using time-series categorical data, has been successfully applied to predict various clinical outcomes[23]. Despite the potentially high expressive power of deep learning models, deep learning models using time-series EHR data to predict drug-resistant bacteria, particularly for MRSA, are limited[24].

We created a deep learning-based prediction model using PyTorch_EHR for positive MRSA culture using big time-series EHR data from a local hospital system and compared it to the traditional machine learning approaches and clinician's decisions of empirical therapy against MRSA. We also evaluated the model's generalizability using external EHR data from a different region of the United States. PyTorch_EHR outperforms logistic regression (LR) and light gradient boost machine (LGBM) models in accuracy (Area Under Receiver Operating Characteristic Curve [AUROC] $^{PyTorch\_EHR}$ = 0.911, $AUROC^{LR}$ = 0.857, $AUROC^{LGBM}$ = 0.892). External dataset from the Medical Information Mart for Intensive Care (MIMIC)-IV validates its superior accuracy ($AUROC^{PyTorch\_EHR}$ = 0.859, $AUROC^{LR}$ = 0.816, $AUROC^{LGBM}$ = 0.838). Our model effectively stratifies patients into high-, medium-, and low-risk categories, potentially optimizing antimicrobial therapy and reducing unnecessary MRSA-specific antimicrobials.

## Results

### Patient characteristics
A total of 26,233 and 152,979 patients who met our selection criteria, as described under Methods, were identified from the Memorial Hermann Hospital System (MHHS) and Medical Information Mart for Intensive Care (MIMIC)-IV databases, respectively. Those patients had 56,233 and 393,713 index culture events over time in MHHS and MIMIC-IV datasets. The aggregated patient characteristics are described in Table 1. Some patients were classified into MRSA and non-MRSA groups when they had both MRSA and non-MRSA events at different index time. Patient features were used once if the patient had two or more events in the same group. Demographic features at the time of index culture were used to describe the characteristics when patients were classified more than twice into one group. Overall, the MRSA group had a higher number of intensive care unit (ICU) admissions (MHHS: 4.3% vs. 0.7%, MIMIC-IV: 31.7% vs. 16.7%) and emergency department (ED) patients (MHHS: 66.4% vs. 13.3%, MIMIC-IV: 51.3% vs 35.0%). As MIMIC-IV was originally developed based on an ICU database, the MIMIC-IV dataset included a higher number of ICU patients. Intermediate unit (IMU) status was not included in the

MIMIC-IV data. Table 2 summarizes types of antibiotics and cultures before index time. Vancomycin was the most commonly used antibiotic, followed by cefepime in the MHHS dataset, whereas ceftriaxone was the second most commonly used antibiotic in the MIMIC-IV dataset. As expected, given the origin of the EHRs (MHHS from Houston and MIMIC-IV from Boston), the MHHS dataset had more Hispanic patients compared to MIMIC-IV (10.5–10.6% vs. 3.6–3.9%). Across groups, Caucasian was the most common race, and 55–65 years was the most common age group. Gender was equally distributed in all groups. Blood and urine cultures were other common cultures taken during the study periods.

### Types of infection and other pathogens
Table 3 summarizes the bacteria and diagnostic codes identified within the event periods. *S. aureus* were the most common bacteria in MRSA groups, whereas *E. coli* was the most common in the non-MRSA group. Bacteremia (MHHS: 6.7% vs. 2.1%, MIMIC-IV: 8.6% vs. 1.9%) and skin soft tissue infection (MHHS: 24.8% vs. 5.6%, MIMIC-IV: 13.2% vs. 2.6%) were more common in MRSA groups.

### Model prediction
Table 4 shows the prediction accuracy of the models. For the MHHS dataset, the deep learning model PyTorch_EHR exhibited the highest Area Under Curve of Receiver Operating Characteristics (AUROC) of 0.911 [0.900 – 0.916] (see ROC curve in Supplementary Fig. 5-1) compared to other machine learning models (logistic regression [LR]: 0.857 [0.849–0.865] and light gradient boost machine [LGBM]: 0.892 [0.885–0.899]). Similar results were obtained for the MIMIC-IV dataset (PyTorch_EHR: 0.859 [0.849–0.869], LR: 0.816 [0.804–0.828], and LGBM: 0.838 [0.823–0.849]; see ROC curve in Supplementary Fig. 5-2). We also evaluated the AUROC in each patient group with a specific diagnosis during the event. Although the AUROC decreased by 0.50–0.10, we had acceptable accuracy in each infection in the MHHS dataset. We also evaluated confusion matrices based on our model's high-risk and low-risk predictions (see Supplementary Table 4). In high-risk groups, Pytorch_EHR showed a specificity of 95.0% and 99.0%, and a sensitivity of 48.1% and 19.3% in MHHS and MIMIC-IV datasets, respectively, whereas LGBM showed a specificity of 95.0% and 99.0%, and a sensitivity of 44.5% and 14.9%. In low-risk groups, Pytorch_EHR had a sensitivity of 95.0% and 90.0% and a specificity of 62.9% and 58.7% in MHHS and MIMIC-IV datasets, respectively, whereas LGBM showed a sensitivity of 95.0% and 90% and a specificity of 62.8% and 57.2%.

Given the imbalanced distributions of positive events in both datasets, for high-risk patients, positive predictive values (PPV) were relatively low: 65.6% and 22.4% for Pytorch_EHR and 63.6% and 17.5% for LGBM in MHHS and MIMIC-IV datasets, respectively. However, negative predictive values (NPV) were high: 90.3% and 98.9% for Pytorch_EHR and 89.7% and 98.8% for LGBM in MHHS and MIMIC-IV datasets, respectively. For low-risk patients, PPV was low: 37.6% and 3.0% for Pytorch_EHR and 33.5% and 2.9% for LGBM in MHHS and MIMIC-IV datasets, respectively. However, NPV were particularly high: 98.6% and 99.8% for Pytorch_EHR and 98.5% and 99.8% for LGBM in MHHS and MIMIC-IV datasets, respectively.

Fig. 1 shows the cumulative incidence curve of MRSA-positive cultures over two weeks from the index culture. In both datasets, our model clearly differentiated the patients with high and low risks of MRSA-positive cultures. The cumulative incidence of MRSA-positive cultures in the MRSA group in the MHHS dataset was 61.2%, whereas the incidence in the MIMIC-IV dataset was approximately 18.2%. The low incidence in MIMIC-IV despite a high risk was likely due to the overall incidence of positive MRSA cultures in the MIMIC-IV dataset.

AUROC curves over multiple index events were evaluated in MHHS and MIMIC-IV test datasets. (See Supplementary Fig. 10) When evaluated on patients with only the first event in MHHS dataset, LGBM

**Table 1 | Characteristics of Patients with and without Positive MRSA Cultures**

| | Memorial Hermann Hospital System | | MIMIC-IV | |
|---|---|---|---|---|
| | MRSA group N = 8164 [a] | Non-MRSA group N = 22,393 [a] | MRSA group N = 4107 [a] | Non-MRSA group N = 152,006[a] |
| Hospital location[b]: Top 5 | N (%) | N (%) | N (%) | N (%) |
| ICU | 354 (4.3%) | 166 (0.7%) | 1302 (31.7%) | 25,319 (16.7%) |
| IMU | 283 (3.5%) | 124 (0.6%) | – | – |
| ED | 5421 (66.4%) | 2981 (13.3%) | 2106 (51.3%) | 53,185 (35.0%) |
| Age | | | | |
| 55–65 | 1965 (24.1%) | 3637 (16.2%) | 943 (23.0%) | 30,196 (19.9%) |
| 65–75 | 1943 (23.8%) | 3857 (17.2%) | 934 (22.7%) | 28,551 (18.8%) |
| 45–55 | 1483 (18.2%) | 2752 (12.3%) | 729 (17.8%) | 23,956 (15.8%) |
| Gender, Male | 4200 (51.4%) | 9897 (44.2%) | 2338 (56.9%) | 86,718 (57.0%) |
| Ethnicity | | | | |
| Hispanic | 866 (10.6%) | 2,358 (10.5%) | 159 (3.9%) | 5400 (3.6%) |
| Non-Hispanic | 5671 (69.5%) | 11,210 (50.1%) | 2810 (68.4%) | 78,559 (51.7%) |
| Unknown | 651 (8.0%) | 1064 (4.8%) | | |
| Race: Top 5 | | | | |
| White | 3123 (38.3%) | 6576 (29.4%) | 2153 (52.4%) | 55,657 (36.6%) |
| African American | 1555 (19.0%) | 2878 (12.9%) | 406 (9.9%) | 12,701 (8.4%) |
| Asian | 103 (1.3%) | 413 (1.8%) | 53 (1.3%) | 3,233 (2.1%) |
| Other | 2174 (26.6%) | 4470 (20.0%) | 281 (6.8%) | 9872 (6.5%) |
| Unknown | 470 (5.8%) | 905 (4.0%) | 185 (4.5%) | 4,991 (3.3%) |
| Primary Language[c] | | | | |
| English | 6314 (77.3%) | 12,927 (57.7%) | 2699 (65.7%) | 74,779 (49.2%) |
| Spanish | 248 (3.0%) | 743 (3.3%) | – | – |
| Unknown | 732 (9.0%) | 1260 (5.6%) | | |
| Selected Comorbidities[d] | | | | |
| Cerebrovascular accident | 918 (11.2%) | 4700 (21.0%) | 328 (8.0%) | 14,360 (9.4%) |
| Congestive Heart Failure | 1408 (17.2%) | 7021 (31.4%) | 1034 (25.2%) | 37,401 (24.6%) |
| Chronic pulmonary diseases | 1041 (12.8%) | 5779 (25.8%) | 1156 (28.1%) | 51,492 (33.9%) |
| Cirrhosis | 149 (1.8%) | 1175 (5.2%) | 184 (4.5%) | 13,126 (8.6%) |
| Chronic Kidney Disease | 1438 (17.6%) | 8049 (35.9%) | 1081 (26.3%) | 41,702 (27.4%) |
| Hypertension | 3008 (36.8%) | 17,429 (77.8%) | 2228 (54.2%) | 106,436 (70.0%) |
| Diabetes Mellitus | 1790 (21.9%) | 9225 (41.2%) | 1411 (34.4%) | 52,323 (34.4%) |
| Malignancy | 477 (5.8%) | 3380 (15.1%) | 767 (18.7%) | 45,128 (29.7%) |
| HIV/AIDS | 136 (1.7%) | 541 (2.4%) | 56 (1.4%) | 2829 (1.9%) |

*AIDS* Acquired Immunodeficiency Syndrome, *ED* Emergency Department, *HIV* Human Immunodeficiency Virus, *ICU* Intensive Care Unit, *IMU* Intermediate Unit, *MIMIC* Medical Information Mart for Intensive Care, *MRSA* Methicillin-Resistant *Staphylococcus aureus*, *N* Number.

[a]*Patients who had positive cultures for both MRSA and non-MRSA at different prediction periods were included in both groups. The number is unique number of patients in the group.*

[b]Patients who had multiple encounters, orders, or documentation during study periods were counted separately unless documented on the same date.

[c]If multiple languages were documented, they were counted separately.

[d]Since some patients had multiple encounters during the study period, diagnostic codes are summarized based on patient levels rather than encounter levels.

model performance was better than that of PyTorch_EHR and LR models. However, when evaluated on patients who had repeated events, i.e., a longer duration of observation in the dataset, PyTorch_EHR model performance improved significantly and sustained superiority against the LR and LGBM models. Similar results were obtained for the MIMIC-IV dataset, with a longer duration of observation providing better performance in the PyTorch_EHR model.

**Potential clinical impact**

Table 5 summarizes the potential clinical impact of the PyTorch_EHR model. In patients predicted as low risk, our model exhibited NPV of 98.6% and 99.8% in MHHS and MIMIC-IV datasets, respectively. In addition, among those low-risk patients who had true negative results, MRSA-specific antimicrobials were given by treating clinicians in 21.6% (1505/6975) and 2.3% (1069/45,533) of events, which translated to 7949 and 1397 doses of MRSA-specific antimicrobials in MHHS and MIMIC-IV, respectively. The main antimicrobials used for those patients were

vancomycin (6833 and 1254 doses in MHHS and MIMIC-IV, respectively), followed by linezolid (852 and 88 doses) and daptomycin (264 and 55 doses). Further, 1.4% (98/6,975) and 0.2% (108/45,533) events were false negatives in our model. Among them, only 0.3% (23/6,975) and 0.04% (27/45,533) events received MRSA-specific antimicrobials, which could be missed by our model.

In high-risk patients, our model exhibited PPV of 65.6% and 22.4% in MHHS and MIMIC-IV datasets, respectively (Supplementary Table 4). The model predicted 12% (1437/11,922) and 1.2% (957/78,548) of events as high risk. Among high-risk groups, patients did not receive any MRSA-specific antimicrobials in 34.6% (497/1437) and 19.7% (189/957) of events in MHHS and MIMIC-IV datasets, respectively. On the contrary, with our model's high-risk prediction, 15.8% (227/1437) and 71.1% (671/957) events may receive unnecessary MRSA-specific antimicrobials (potential harm from our model).

Finally, we evaluated the performance of our model in patients who had MRSA bacteremia. As summarized in Tables 5, 31.8% (457/1437) and

**Table 2 | Types of antibiotics and cultures which patients had before Index Time**

|  | Memorial Hermann Hospital System | MIMIC-IV | | |
|  | MRSA group N = 8164 [a] | Non-MRSA group N = 22,393 [a] | MRSA group N = 4107 [a] | Non-MRSA group N = 152,006 [a] |
|---|---|---|---|---|
| *Total Antibiotics Given in The Group (days)[b]: Top 5* |  |  |  |  |
| Vancomycin | 28,153 | 21,185 | 7647 | 75,879 |
| Cefepime | 19,486 | 14,687 | 3590 | 42,530 |
| Meropenem | 11,960 | 9542 | 1744 | 19,448 |
| Ceftriaxone | 10,674 | 8259 | 2879 | 49,130 |
| Piperacillin-tazobactam | 10,473 | 8227 | 2224 | 25,558 |
| *Total Antibiotics Route Given in The Group (days)[b]: Top 5* |  |  |  |  |
| IV | 71,749 | 54,986 | 16,160 | 191,633 |
| PO | 13,446 | 10,317 | 10,498 | 167,895 |
| Enteric tube | 8275 | 6703 | – | – |
| Inhalation | 561 | 504 | 0 | 0 |
| Ophthalmic | 138 | 121 | 2 | 100 |
| *Total Culture Type Obtained in The Group [b]: Top 5* |  |  |  |  |
| Blood | 14,741 | 33,907 | 17,557 | 128,537 |
| Urine | 6263 | 29,840 | 12,402 | 215,712 |
| Wound | 4769 | 7523 | 4424 | 11,798 |
| Anaerobic | 2150 | 5966 | 5603 | 13,214 |
| Body Fluid/Tissue | 1963 | 6443 | 1946 | 13,974 |

IV Intravenous, PO Per Os, MIMIC Medical Information Mart for Intensive Care, MRSA Methicillin-Resistant Staphylococcus aureus, N Number.
[a]Patients who had positive cultures for both MRSA and non-MRSA at different prediction periods were included in both groups. The number is unique number of patients in the group.
[b]Patients who had multiple encounters, orders, or documentation during study periods were counted separately unless documented on the same date.

**Table 3 | Name of bacteria identified from cultures and types of infection based on ICD codes**

|  | Memorial Hermann Hospital System | MIMIC-IV | | |
|  | MRSA group N = 9773[a] | Non-MRSA group N = 48,461[a] | MRSA group N = 5789[a] | Non-MRSA group N = 387,924[a] |
|---|---|---|---|---|
| *Bacteria Name[b] 5 Common Bacteria* | N (%) | N (%) | N (%) | N (%) |
| S. aureus | 9773 (100%) | 2117 (4.4%) | 5789 (100%) | 10,724 (2.8%) |
| Enterococcus spp. | 848 (8.7%) | 3745 (7.7%) | 360 (6.2%) | 10,140 (2.6%) |
| E. coli | 569 (5.8%) | 8028 (16.6%) | 287 (5.0%) | 27,403 (7.1%) |
| K. pneumoniae | 373 (3.8%) | 2258 (4.7%) | 213 (3.7%) | 7619 (2.0%) |
| P. aeruginosa | 783 (8.0%) | 2224 (4.6%) | 463 (8.0%) | 5753 (1.5%) |
| *Types of Infections[b]* |  |  |  |  |
| Sepsis | 1965 (20.1%) | 5621 (11.6%) | 411 (7.1%) | 8007 (2.6%) |
| Pneumonia | 1130 (11.6%) | 3882 (8.0%) | 675 (11.2%) | 14,669 (3.8%) |
| Bacteremia | 657 (6.7%) | 1045 (2.1%) | 497 (8.6%) | 7531 (1.9%) |
| Skin Soft Tissue Infection | 2420 (24.8%) | 2721 (5.6%) | 762 (13.2%) | 10,106 (2.6%) |
| UTI | 1151 (11.2%) | 12,545 (25.9%) | 652 (11.2%) | 20,929 (5.3%) |

MIMIC Medical Information Mart for Intensive Care, MRSA Methicillin-Resistant *Staphylococcus aureus*, N Number, UTI Urinary Tract Infection
[a]The number represents the number of events during two-week window. Non-MRSA group includes events with negative cultures.
[b]Patients who had multiple bacteria or types of infections were counted separately unless documented on the same index periods.

7.3% (70/957) of high-risk events in MHHS and MIMIC-IV datasets, respectively, had MRSA bacteremia. These rates were much higher than the rates in low-risk events in MHHS (0.5%; 32/6975) and MIMIC-IV (0.04%; 35/48,455). Based on these findings, high-risk group had 69.3 and 101.2 higher relative risk of MRSA bacteremia compared to low-risk patient group. In addition, our model identified 58.0% (265/457) and 50.0% (35/70) of high-risk patients with true MRSA bacteremia did not receive MRSA-specific antimicrobials, considered "optimal" antibiotics for MRSA bacteremia, within 12 h of the index cultures.

These results were also evaluated in other models and any MRSA antimicrobials (see Supplementary Table 5). Overall, PyTorch_EHR

model exhibited higher net-benefits against treating clinician's decisions compared to LGBM and LR models, except for MRSA bacteremia in MIMIC-IV dataset. LGBM model provided better net benefit compared to PyTorch_EHR model (18 vs. 10 MRSA bacteremia cases may receive early MRSA antimicrobials, respectively.)

**Feature importance**

We obtained the contribution scores for positive MRSA cultures in the datasets. Supplementary Fig. 7 shows the top 14 median contribution scores of admission diagnoses in our model for MHHS data. Interestingly, our model identified multiple diagnoses often related to MRSA

**Table 4 | Outcome of Models in Overall and Subgroup Analyses**

| | | Memorial Hermann Hospital System AUROC Average (CI) | MIMIC-IV AUROC Average (CI) |
|---|---|---|---|
| | PyTorch_EHR | 0.911 (0.900–0.916)* | 0.859 (0.849–0.869)* |
| Overall Prediction | PyTorch_EHR Pre-Trained | – | 0.860 (0.850–0.871)* |
| | LR | 0.857 (0.849–0.865) | 0.816 (0.804–0.828) |
| | LGBM | 0.892 (0.885 – 0.899) | 0.838 (0.823–0.849) |
| Subgroups Analysis | | | |
| | PyTorch_EHR | 0.864 (0.846–0.882)* | 0.789 (0.740–0.840) |
| Sepsis | PyTorch_EHR Pre-Trained | – | 0.781 (0.734–0.828) |
| | PyTorch_EHR | 0.879 (0.842–0.915) | 0.797 (0.755–0.840) |
| Bacteremia | PyTorch_EHR Pre-Trained | – | 0.809 (0.770–0.848) |
| | PyTorch_EHR | 0.872 (0.849–0.894)* | 0.783 (0.743–0.823) |
| Pneumonia | PyTorch_EHR Pre-Trained | – | 0.769 (0.730–0.807) |
| | PyTorch_EHR | 0.804 (0.778–0.831) | 0.819 (0.783–0.856) |
| Skin Soft Tissue Infections | PyTorch_EHR Pre-Trained | - | 0.811 (0.775–0.847) |

CI Confidence Interval, LGBM Light Gradient-Boosting Machine, LR Logistic Regression, MIMIC Medical Information Mart for Intensive Care.
*Statistically significantly better compared to light GBM.

infections, such as cutaneous abscesses or boils. Supplementary Fig. 8 shows the top 10 overall contribution scores for antimicrobial exposures before the index time in the datasets. Some common antibiotics had high scores in both datasets, but it was difficult to interpret the scores clinically.

We also present individual feature importance as a bar graph for an example patient among the patients we visualized (see Supplementary Fig. 9). The patient is female and between 45 – 54 years of age, with multiple underlying comorbidities listed on admission two days (−2 days) before the index culture (blood culture on index date). Our model identified a risk score of 0.541 (predicted as a positive patient). After the patient was admitted to the hospital, vancomycin and meropenem were initiated, and a blood culture was ordered. Subsequently, cultures identified MRSA over two weeks.

## Discussion
In this study, our deep learning-based MRSA-predictive model exhibited better performance compared to other machine learning models in real-world MHHS and MIMIC-IV datasets. Traditional Machine learning, especially LGBM, also provided a great performance in predicting MRSA-positive culture. However, PyTorch_EHR model had better overall AUCROC and showed better potential clinical impact in majority of datasets. PyTorch_EHR model successfully "learned" patient-specific features, especially with time sequence events, to provide personalized risks of positive MRSA cultures over two weeks from index time. The model maintained better predictions even after transferring from the MHHS dataset to the MIMIC-IV dataset and tolerated the significantly imbalanced outcomes in the MIMIC-IV dataset. Compared to other existing models, our model successfully predicted positive MRSA cultures not only on the index day but also over two weeks from the index day. (see Fig. 2) This prediction window is better aligned with the daily clinical practice of physicians since physicians decide on empiric antibiotic therapy to treat MRSA, such as intravenous vancomycin, not only for the culture of index day but also any subsequent cultures that may be related to the episode of infection after initiation of therapy. We decided to use a two-week window in this project as the majority of infections after admission are diagnosed within the time periods. The incidence curve successfully captured any events within two weeks. Our deep learning model readily accepts the time sequence of the events in the patient history, which we believe is more consistent with the physician's assessment in clinical practice. In addition, our model showed that accuracy improves when time-series data are used, and patients have a longer duration of observation before the index time. (see Supplementary Fig. 10) We also tested the model in different types of infection posing various MRSA risks, such as sepsis, bacteremia, and pneumonia. Although there were some decreases in the AUROC, high performance were maintained, which supports the use of this single model for multiple types of infections. Finally, our model could benefit clinical practice by reducing the number of antimicrobials used in low-risk patients and providing optimal MRSA antimicrobials when the model predicts high risk, including bacteremia. Although the difference in AUCROC was small between PyTorch_EHR and LGBM, the actual difference of possible net-benefit is substantial, especially in MHHS datasets which had high prevalence rates of positive MRSA cultures.

Personalized medicine is of great interest in medical fields. Many studies on personalized medicine focus more on genetic-based predictions rather than clinical data from EHRs[25]. EHR data have become a rich source of real-world data and provide invaluable information. Even without genetic data, we believe EHR data can be a useful source for deep learning models to achieve personalized medicine in multiple clinical settings. Furthermore, compared to traditional machine learning models, deep learning can easily integrate time-sequence data as inputs into the model, which provides significant advantages for outcome predictions requiring sequential event inputs. Although PyTorch_EHR only uses categorical data from EHRs, this model provides high performance with the advantages of relatively simpler preprocessing steps and flexible variable selections for input. This allows us to preserve model transferability and generalizability across different data sources.

Since MRSA emerged, multiple predictive models for risk factors for MRSA infections have been proposed. The models have differing degrees of accuracy but often focus on a certain type of infection, such as pneumonia, to achieve and simplify the risk factors and models. Rhodes et al. used a machine learning model to predict community-acquired MRSA pneumonia[26]. Although the time frame and patient population differed from our study, their model achieved an AUROC of 0.775, which was lower than ours. Additionally, some risk factor-based models rely heavily on certain tests, such as the nasal MRSA PCR test from nare[27], which hampers the model's generalizability due to limitations in the tests' availability and applicability to other types of infections. Also, some of the results may not be available when starting antibiotics, which limits the usability of models in hospitals. In contrast, our model carries a significant advantage since the model can

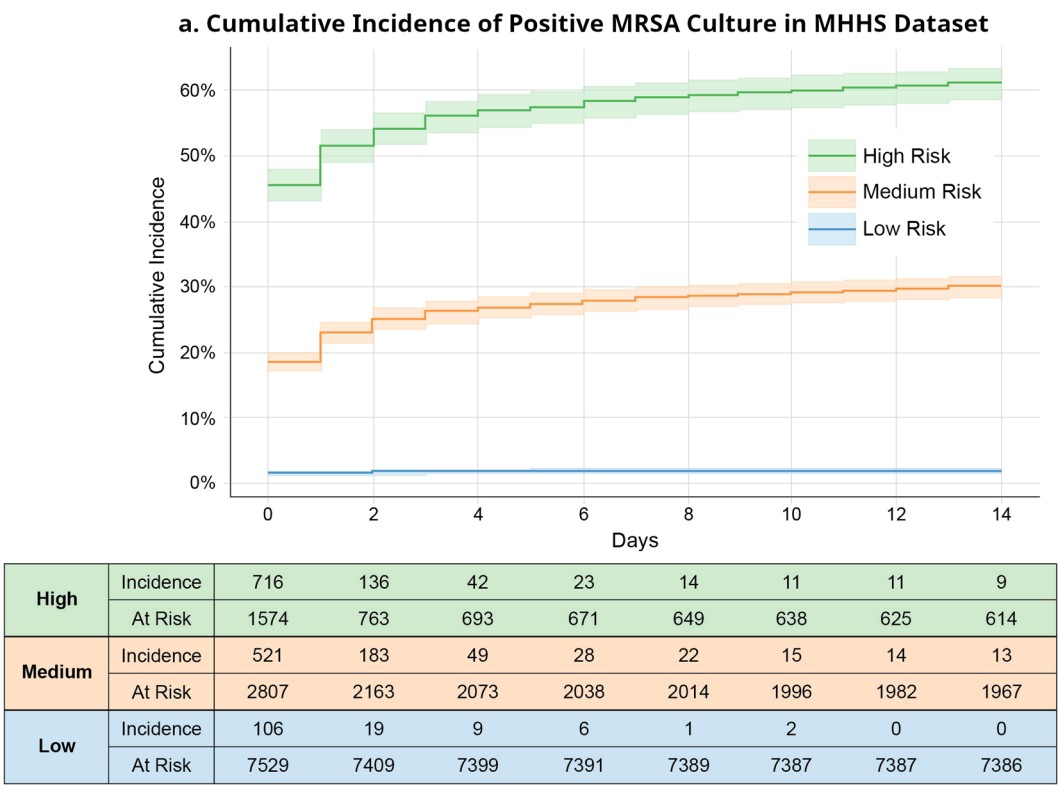

## a. Cumulative Incidence of Positive MRSA Culture in MHHS Dataset

| | | | | | | | | | |
|---|---|---|---|---|---|---|---|---|---|
| **High** | Incidence | 716 | 136 | 42 | 23 | 14 | 11 | 11 | 9 |
| | At Risk | 1574 | 763 | 693 | 671 | 649 | 638 | 625 | 614 |
| **Medium** | Incidence | 521 | 183 | 49 | 28 | 22 | 15 | 14 | 13 |
| | At Risk | 2807 | 2163 | 2073 | 2038 | 2014 | 1996 | 1982 | 1967 |
| **Low** | Incidence | 106 | 19 | 9 | 6 | 1 | 2 | 0 | 0 |
| | At Risk | 7529 | 7409 | 7399 | 7391 | 7389 | 7387 | 7387 | 7386 |

## b. Cumulative Incidence of Positive MRSA Culture in MIMIC-IV Dataset

| | | | | | | | | | |
|---|---|---|---|---|---|---|---|---|---|
| **High** | Incidence | 155 | 12 | 3 | 3 | 2 | 1 | 0 | 2 |
| | At Risk | 974 | 810 | 805 | 803 | 800 | 799 | 799 | 79 |
| **Medium** | Incidence | 290 | 109 | 43 | 30 | 18 | 11 | 7 | 12 |
| | At Risk | 25805 | 25439 | 25383 | 25351 | 25324 | 25308 | 25303 | 25288 |
| **Low** | Incidence | 45 | 31 | 17 | 3 | 5 | 10 | 7 | 1 |
| | At Risk | 47755 | 47686 | 47670 | 47662 | 47659 | 47652 | 47643 | 47637 |

**Fig. 1 | Cumulative Incidence Curve of Positive MRSA Over Two Weeks in the MHHS and MIMIC-IV Datasets. a** and **b** show cumulative incidence of MRSA cultures in Memorial Hermann Hospital System (MHHS) and Medical Information Mart for Intensive Care (MIMIC)-IV datasets, respectively. Both figures were generated based on the risk predicted by our model in test datasets. Given the significant imbalance in the MIMIC-IV dataset, even high-risk patients achieved 20% positivity compared to the MHHS dataset. In contrast, the low-risk patient group had fewer false negatives. The shaded area in the graph represents the 95% confidence intervals. MHHS Memorial Hermann Hospital System, MIMIC Medical Information Mart for Intensive Care, MRSA Methicillin Resistant *Staphylococcus aureus*.

**Table 5 | Potential Clinical Impact of the PyTorch_EHR Model**

| MHHS Data (PyTorch_EHR) | | | | | | |
| --- | --- | --- | --- | --- | --- | --- |
| **Model Predictions** | | MRSA Cx | **Treating Clinician's decision** | **Cases** | **Potential Benefit and Harm of Model** | **Overall Potential Benefit** |
| High Risk 1437 PPV: 65.6 | True Positive 943 (65.6%) | + | Empirically Treat | 446/1437 (31.0%) | NA | **474 cases may receive early MRSA Abx** **1278 cases may avoid unnecessary MRSA Abx** |
| | | **+** | **Not Empirically Treat** | **497/1437 (34.6%)** | **497 cases may get early MRSA Abx.** | |
| | False Positive 494 (35.4%) | – | Empirically Treat | 267/1,437 (18.5%) | NA | |
| | | – | *Not Empirically Treat* | *227/1,437 (15.8%)* | *227 cases may receive unnecessary MRSA Abx.* | |
| Low Risk 6975 NPV: 98.6 | True Negative 6877 (98.7%) | – | **Empirically Treat** | **1505/6975 (21.6%)** | **1505 cases may avoid unnecessary MRSA Abx.** | |
| | | – | Not Empirically Treat | 5372/6975 (77.0%) | NA | |
| | False Negative 98 (1.4%) | + | *Empirically Treat* | *23/6,975 (0.3%)* | *23 cases may delay MRSA Abx* | |
| | | + | Not Empirically Treat | 75/6,975 (1.1%) | NA | |
| MRSA Bacteremia 457 | True Positive 434 (95.0%) | + | Empirically Treat | 169/457 (37.0%) | NA | **250 MRSA bacteremia cases may start early MRSA Abx** |
| | | **+** | **Not Empirically Treat** | **265/457 (58.0%)** | **265 cases may start early MRSA Abx** | |
| | False Negative 23 (5.0%) | + | *Empirically Treat* | *15/457 (3.3%)* | *15 cases may delay MRSA Abx* | |
| | | + | Not Empirically Treat | 17/457 (3.7%) | NA | |
| MIMIC-IV (PyTorch_EHR) | | | | | | |
| Model Predictions | | MRSA Cx | Treating Clinician's decision | Cases | Potential Benefit and Harm of Model | Overall Potential Benefit |
| High Risk 957 PPV: 22.4 | True Positive 214 (21.1%) | + | Empirically Treat | 25/957 (2.6%) | NA | **162 cases may receive early MRSA Abx** **398 cases may avoid unnecessary MRSA Abx** |
| | | **+** | **Not Empirically Treat** | **189/957 (19.7%)** | **189 cases may get early MRSA Abx** | |
| | False Positive 743 (77.6%) | – | Empirically Treat | 72/957 (7.5%) | NA | |
| | | – | *Not Empirically Treat* | *671/957 (70.1%)* | *671 cases may receive unnecessary MRSA Abx* | |
| Low Risk 45,533 NPV: 99.8 | True Negative 45,425 (99.8%) | – | **Empirically Treat** | **1069/ 45,533 (2.3%)** | **1069 cases may avoid unnecessary MRSA Abx** | |
| | | – | Not Empirically Treat | 44,356/ 45,533 (97.4%) | NA | |
| | False Negative 108 (0.2%) | + | *Empirically Treat* | *27/45,533 (0.04%)* | *27 cases may delay MRSA Abx* | |
| | | + | Not Empirically Treat | 81/45,533 (0.17%) | NA | |
| MRSA Bacteremia 70 | True Positive 35 (50.0%) | + | Empirically Treat | 8/70 (11.4%) | NA | **10 MRSA bacteremia cases may receive early MRSA Abx** |
| | | **+** | **Not Empirically Treat** | **27/70 (38.6%)** | **27 cases may start early MRSA Abx** | |
| | False Negative 35 (50.0%) | + | *Empirically Treat* | *17/70 (24.3%)* | *17 cases may delay MRSA Abx* | |
| | | + | Not Empirically Treat | 18/70 (25.7%) | NA | |

*Abx* Antibiotics, *Cx* Culture, *MHHS* Memorial Hermann Hospital System, *MIMIC* Medical Information Mart for Intensive Care, *MRSA* Methicillin-Resistant *Staphylococcus aureus*, *NPV* Negative Predictive Value, *PPV* Positive Predictive Value.
This table summarizes the potential clinical benefits or harms of our model compared to the treating clinician's decisions. The numbers obtained are based on antimicrobials possessing MRSA-specific activities. Bolded lines indicate the potential benefits with our model, and italic lines indicate the potential harms with our model. In each dataset, the overall net benefit outweighed the harms with our model even compared to the treating clinician's decision, i.e. overall 1669 cases and 424 cases potentially get benefit from our model in MHHS and MIMIC-IV cohort, respectively.

take widely available data from EHRs and predict the outcomes even with some missing certain tests. Our model can be used not only for treatment decisions but also for infection prevention to isolate the patients in high-risk groups before culture results, although the utility of contact precaution is still controversial. Our model used a two-week time window to provide more meaningful predictions in clinical settings. Some predictive models only predict the index culture rather than overall risks[21]. To be applied in a clinical setting, predicting over a two-week window can be more impactful for clinicians when they choose antimicrobial therapy at the time of the initiation. The cumulative incidence curves based on our model prediction clearly differentiated the high-risk and low-risk patients. The majority of patients had positive MRSA cultures on the index day, but approximately 15% of high-risk patients had positive cultures after the index day, which could be missed if we only predicted the positivity of the index culture. Currently, our model only predicts the positivity of cultures regardless of the source of cultures, which allows simplification of the data

processing and model structure. However, providing more informative predictions, such as source of cultures, may allow physicians to decide finer selection of antimicrobial therapies. For example, when the model predicts only wound culture is positive for MRSA in stable cellulitis patients, oral antimicrobials, such as sulfamethoxazole-trimethoprim, may be adequate for the therapy.

We evaluated the potential impact of our PyTorch_EHR model in a clinical setting. MRSA-specific and any MRSA antibiotics were used to evaluate the impacts. Although linezolid and daptomycin can be used for vancomycin-resistant enterococci (VRE), the low-risk groups had positive VRE in 6 cases in MHHS test datasets and 32 cases in MIMIC-IV test datasets. The model identified a large number of potentially avoidable antimicrobials targeting MRSA used in low-risk patients (7949 and 1397 doses in MHHS and MIMIC-IV, respectively). Our model only "missed" a small number of patients (0.3% and 0.04% in MHHS and MIMIC-IV, respectively). When evaluating overall performance, our model potentially provides benefits in 1752 cases and 560 cases in

## a. Schematic Structure of MRSA-Positive Culture Deep Learning-Based Prediction Model

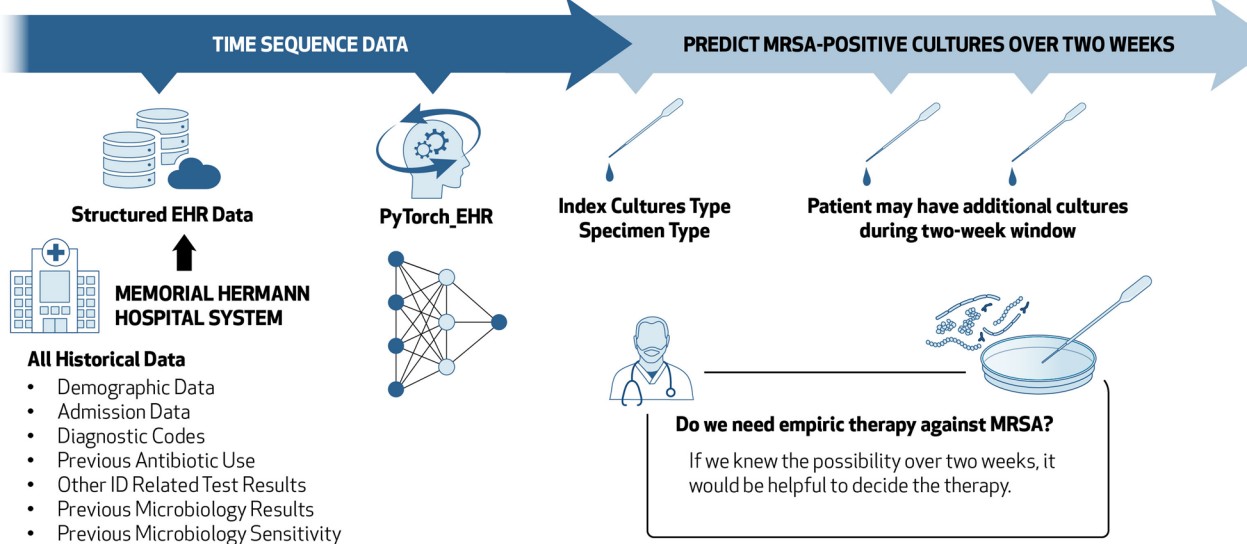

## b. Schematic Explanation of Patients Who Had Multiple Cultures Over Multiple Encounters

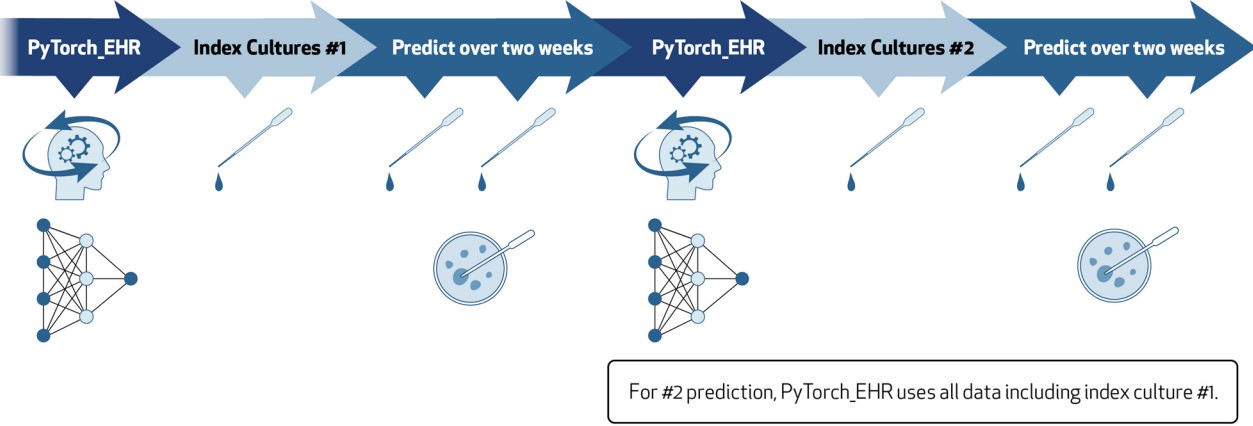

**Fig. 2 | Schematic Structure of Deep Learning-Based Prediction Model for MRSA-Positive Cultures. a** summarizes the overall structure of the model used to predict MRSA-positive cultures over a two-week period from the index culture. Our model integrates multiple structural data tables from Electronic Health Records (EHRs) as time-sequenced data prior to the index time. A deep learning-based model (PyTorch_EHR) is employed to predict MRSA-positive cultures over two weeks from the index time. **b** describes scenarios where patients experience multiple events over time. EHR: Electronic Health Records, MRSA: Methicillin-Resistant *Staphylococcus aureus*.

MHHS and MIMIC-IV datasets. The high-risk patient population had a significantly high relative risk ratio for MRSA bacteremia. This indicates that our model predicts not only the positivity of MRSA culture but also the severity of MRSA infections in high-risk patients. Furthermore, although the absolute numbers were small, 58.0% and 38.6% of events with MRSA bacteremia did not receive "optimal" antimicrobials within 12 h of the index time. MRSA bacteremia is one of the most severe infections in the hospital. In our study, although only early antimicrobial therapy and avoidable MRSA-specific antimicrobials were evaluated, early initiation of appropriate antibiotics in critically ill patients improves their outcomes[1] and avoiding unnecessary antimicrobials reduces side effects and potential complications from those antimicrobials, such as *Clostridioides difficile* infection. We believe potential benefits can be larger in clinical settings.

One of the challenges of deep learning models is their explainability. Interestingly, our model successfully identified clinically important features. Although there was variability among patients, our model successfully identified MRSA-related admission diagnosis. Previous antimicrobial exposures were also visualized in the population.

However, the results were difficult to interpret clinically. We also visualized the factors contributing to the model predictions at an individual level (see Supplementary Fig. 9). Since the model uses the time sequence without dichotomizing the time frame with an arbitrary cutoff, i.e., positive MRSA culture within 90 days, the contribution weight can be different depending on the patient and the timing of events. Although some of the factors seem associated with MRSA infection, those highly contributed events are not necessarily directly associated with the predictions of MRSA. The inputs could surrogate other underlying events. Caution is required to interpret the feature importance as those outputs may not be traditional risk factors we use in clinical settings.

This study has limitations. First, due to the nature of retrospective studies, potential biases are inevitable, and its findings should be confirmed in prospective studies. In addition, although the datasets we used are from hospitals in two distinct regions of the United States, the model should be validated in other patient populations and high-risk populations, such as immunocompromised patients. Second, this model predicts positive MRSA cultures rather than infections. Since

some patients can have MRSA infections without positive cultures, the model should be used cautiously when there are significant concerns about MRSA when initiating antibiotics. We also included analysis for patients with MRSA bacteremia, which is usually considered a true infection. The potential clinical benefits were consistent in this cohort. Third, potential clinical impacts by our model were evaluated based on the clinician's antibiotic prescriptions and final culture results. We used MRSA-specific antimicrobials and any MRSA antimicrobials to evaluate the different clinical scenarios. Our model consistently showed the benefits in both settings. Particularly in the evaluation of MRSA-specific antimicrobials, although vancomycin is often used to target MRSA, linezolid and daptomycin can be used for other potential pathogens, such as VRE. Although those were minor cases in the datasets, there could be uncommon situations where those antibiotics were used for other purposes. Fourth, although we included multiple variables in the model, several important variables as known MRSA risk factors, such as residence in a long-term care facility, were not included. Furthermore, vital signs or other basic laboratory results were not included in this model. Those can be considered in future studies. Finally, although we showed the generalizability of the model in this study, the transferability of the model needs to be addressed to use the deep-learning model widely.

In summary, our deep learning-based predictive model successfully predicted positive MRSA culture over two weeks from index culture. Our study revealed model superiority against other traditional machine learning models in both MHHS and MIMIC-IV datasets with high performance, even in significantly imbalanced datasets and some subgroup analyses. The model can be widely applied to various types of infections. Compared to the treating physician's decision, our model could provide potential benefits, reducing unnecessary MRSA antimicrobial use and optimizing antimicrobial therapy. Considering the performance of our model in the datasets, the model likely provides more clinical benefits in populations with a high prevalence of MRSA infections. Studies in high-risk populations, such as immunocompromised patients, and prospective studies are warranted to validate the model.

## Methods

### EHR datasets
Patient data were retrospectively retrieved from two EHR databases: 1) Database at MHHS, Houston, Texas, for model training and comparison to traditional machine learning models and 2) MIMIC-IV v2.1 for external validation. MIMIC-IV is a relational de-identified EHR database containing hospital encounters from a tertiary academic medical center in Boston, Massachusetts[28].

From the MHHS database, EHRs from 1/2018 and 4/2021 were obtained for patients >= 18 years of age, with at least one bacterial culture during the study period. To avoid an imbalanced dataset, we randomly selected 8,164 patients with MRSA-positive cultures and 18,069 patients with other types of cultures, including cultures positive for methicillin-sensitive *S. aureus* (MSSA) and other types of bacteria and negative cultures. Demographic data, admission data, diagnostic and procedure codes, antibiotic administration, other infectious disease-related test results, and previous microbiological data, including the type of cultures, name of bacteria, and all antibiotic sensitivities, were obtained from the database. Microbiology data tables included cultures and other infectious disease tests, such as serologies. To avoid label leakage, we used only results reported by the index time. The laboratory orders were included without results when they were ordered by the index time. For diagnostic and procedure codes, International Classification of Disease (ICD)−9 or ICD-10 codes were used. Since other data tables, such as antibiotics, did not contain standardized codes for medications, free text, such as "vancomycin," was used. Extracted data were cleaned and converted to categorical data to fit the PyTorch_EHR scheme. The admission ward information

was converted to generalized features, such as ED, ICU, and IMU, to later map those locations to MIMIC-IV data.

Similarly, EHRs for all patients with bacterial cultures and >18 years of age were retrieved from the MIMIC-IV database. To validate the generalizability of the model, each data table was mapped with the MHHS data table. Only data mapped with MHHS data were used in the MIMIC-IV dataset. Since the MIMIC-IV dataset aggregated the ICD and procedure codes at each encounter level, only codes reported in the previous encounters were used to avoid label leakage. The microbiology event table was used to identify eligible patient events, and those data were used as part of inputs in our model. Of the total 25,599 *S. aureus*-positive cultures from various sources in the table of MIMIC-IV, 19,605 isolates (76.6%) had been tested for antimicrobial sensitivity for various reasons, including multiple positive cultures with *S. aureus* in a short period and positive wound cultures due to multiple organisms. *S. aureus*-positive cultures within seven days of positive MSSA or MRSA were removed, leaving 519 *S. aureus* isolates, which did not have any recent sensitivity to classify them as MRSA or MSSA. These isolates were classified into the non-MRSA group. The datasets were further divided as 70:10:20 (Supplementary Fig. 1). We used the data for two purposes; 1) to generate a model only trained and tested on MIMIC-IV, and 2) to fine-tune the pre-trained model with the MHHS datasets and test on the MIMIC-IV test dataset. For the results of model predictions and clinical impact, only test datasets of each database were used.

For subgroup analysis in the MHHS dataset, the ICD code was used to identify the patient with that code within the two-week period. Since the MIMIC-IV dataset only provided ICD codes at the encounter levels, we used the encounter to find the patients with the ICD codes within the encounter.

### PyTorch_EHR prediction model scheme
We used the deep learning platform PyTorch_EHR to predict clinical outcomes using categorical data from EHRs. As the majority of MRSA infections or new infections are diagnosed within two weeks, we set a two-week window for the prediction, and any first culture within the window was used as an index culture (Fig. 2). This prediction window allows not only prediction at the time of culture but also cultures obtained after initiation of empiric antibiotics, which is essential for physicians to decide whether to start or continue empiric MRSA antibiotic therapy at the index time. Some patients had multiple cultures over time, including MRSA and non-MRSA cultures. Those patients were included in both MRSA and non-MRSA groups for patient characteristic description, depending on the timing of positive or negative culture the patient had during the window period.

PyTorch_EHR implements an RNN model. We chose the gated recurrent unit (GRU) RNN architecture, which is known for being an efficient sequential deep learning architecture for clinical event predictions (see Supplementary Fig. 1). The source code of this model is publicly available to enable its application and further evaluation by other researchers[29]. In addition to categorical data, PyTorch_EHR handles the time difference between hospital visits for a better temporal representation of patient history to improve accuracy (see Supplementary Fig. 2)[30,31]. We converted the interval to days from visits to accommodate predictions for more acute issues.

For binary classification tasks, we compared our model to two traditional machine learning algorithms, LR[32] and LGBM[33]. We elected LR as the most basic binary classification model and gradient boost machine as powerful and used in multiple classification tasks[34,35]. To keep the temporal relationship between index time and each feature available for those models, we prepared the data to include the number of occurrences of each feature before index time and the distance between the most recent feature occurrence and the index time. (see Supplementary Fig. 3) After preparation of the data, we standardized the numerical values to optimize algorithms. For each model including

RNN, we obtained their optimal hyperparameters using optuna (ver. 2.10.0)[36]. After obtaining the area under the curve (AUC) for each model prediction, we use DeLong test[37] to obtain *p*-values and the 95% confidence intervals of AUC differences between the models to statistically evaluate the significance.

For survival prediction, we used the DeepSurv[38] architecture, replacing the multiple-layer perceptron layers with GRU layers for better sequential information modeling, similar to the way we modelled COVID-19 outcome prediction[23]. Python version 3.9.7, PyTorch version 1.7.1, and Sklearn version 0.24.2 were used in this study.

### Possible clinical impact of our model

To evaluate the potential clinical impact of our model, we filtered high-risk and low-risk patient cohorts based on the prediction of our model. Considering the different prevalences of MRSA-positive cultures in each dataset, we defined different cutoffs for high-risk and low-risk patients. For MHHS dataset, we used the cutoff to obtain a specificity of 95% for high-risk patients and a sensitivity of 95% for low-risk patients. For MIMIC-IV, considering significant imbalanced data, we decided to use a specificity of 99% and a sensitivity of 90%, respectively. All three models used the same cutoff and were evaluated for the model performance. After defining the cohort, we evaluated the number of patients who had positive MRSA cultures and received or did not receive empiric MRSA-related antimicrobial therapy. We used two groups of antimicrobials: MRSA-specific antibiotics and MRSA antibiotics. MRSA-specific antibiotics include vancomycin, daptomycin, linezolid, and telavancin, which are often used in the hospital when empiric therapy is necessary, or bacteremia is suspected. Any MRSA antibiotics include other intravenous and oral antimicrobials, which possess anti-MRSA activity. However, these antimicrobials are also often used for other types of bacteria, such as gram-negative bacteria. We evaluated our model with both groups of antimicrobials. MRSA bacteremia was specifically chosen to define true bacterial infections since positive cultures in other types of culture do not necessarily mean true infections, i.e., some patients may have contamination or colonization in some situations.

### Model interpretation

For the mechanistic interpretation of MRSA predictions, we used the integrated gradient technique[39] to expose the factors contributing to the personalized model predictions. For RNN-based models, we can achieve a patient-level explanation, which shows the contribution scores for each clinical event on each day in the patient trajectory. We also obtained the medians of contribution scores of frequent features in our model to evaluate the overall importance of certain features in the cohort. However, we need to highlight that such contribution scores should be mainly used for patient-level predicted score explanation and not for inferring population-level risk factors/important features as it is different from LR coefficients or LGBM feature importance scores, such as SHAP[40]. To evaluate our RNN-based model explainability, we reviewed the calculated contribution scores for each clinical event in the input of 10 patients. We visualized the contribution score per patient through an institutional Tableau interactive dashboard (Seattle, Washington), where clinicians can navigate different clinical events within various categories and across multiple visits in the patient history.

### Reporting summary

Further information on research design is available in the Nature Portfolio Reporting Summary linked to this article.

## Data availability

MHHS data that support the findings of this study are not openly available due to reasons of sensitivity and are available from the corresponding author upon requests and our institutional IRB approvals. MIMIC-IV data v2.1, used in this study as an external validation, is publicly available after data use agreement on the website. ([https://physionet.org/content/mimiciv/2.1/](https://physionet.org/content/mimiciv/2.1/)).

## Code availability

The Original Pytorch_EHR code and sample codes used in this work are publicly available[29,41].

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

## Acknowledgements
M.N. receives NIH grant (NIH/NIAID R01 AI175699).

## Author contributions
M.N. conducted data cleaning, model training, and wrote manuscripts. B.M. cleaned codes for publication and analyzed data. L.R., Z.X. and D.Z. developed model and revised manuscripts. BSK revised manuscripts and provided insights in model training and evaluation.

## Competing interests
The authors declare no competing interests.

### Ethics approval
This study was approved by Institutional Review Boards (IRBs) at the University of Texas Health Science Center, Houston, Texas, and MHHS (Protocol number: HSC-MS-20-0121).
