## [Peer Review File · Nature Communications]

Deep learning model for personalized prediction of positive MRSA culture using time-series electronic health recordsEditorial Note: This manuscript has been previously reviewed at another journal that is not operating a transparent peer review scheme. This document only contains reviewer comments and rebuttal letters for versions considered at *Nature Communications*.

REVIEWER COMMENTS

Reviewer #1 - no further comments.

Reviewer #3 (Remarks to the Author):

Thank you for asking me to re-review this interesting paper, reporting the comparative performance of models generated using three methods to predict the presence of MRSA infection.

As a clinician scientist, my comments are focused on the clinical elements of the study, which reports relatively complex technical analyses.

I have three main comments/ questions which to my mind could be strengthened in the paper:

1. Why did the authors focus only on MRSA when the clinical problem is more often "should I give antibiotics for a possible severe infection (sepsis)?" rather than "should I give antibiotics for a possible MRSA infection?"
2. The authors present data in Table 4 suggesting possible benefits of using the Deep Learning Model compared with clinical judgment. I was not clear how "clinical judgment" was determined - was this based on the use of certain antibiotics during patients' treatment? If so, how did the authors check if the antibiotic was given for MRSA or something else?
3. Assuming the authors can provide reassurance regarding point 2, if they have adequate data on clinical judgment, could the authors not compare the AUROC curve for Deep Learning Model vs. clinical judgment? Indeed, would this not be a more appropriate comparison (clinically) than with models generated using different methods?

Reviewer #3 (Remarks on code availability):

I have no expertise in the writing or interpretation of code.

REVIEWER COMMENTS

Reviewer #1 - no further comments.

Reviewer #3 (Remarks to the Author):

Thank you for asking me to re-review this interesting paper, reporting the comparative performance of models generated using three methods to predict the presence of MRSA infection.

As a clinician scientist, my comments are focused on the clinical elements of the study, which reports relatively complex technical analyses.

I have three main comments/ questions which to my mind could be strengthened in the paper:

1. Why did the authors focus only on MRSA when the clinical problem is more often "should I give antibiotics for a possible severe infection (sepsis)?" rather than "should I give antibiotics for a possible MRSA infection?"

Thank you so much for your comment. We agree with your point and, indeed, sepsis is one of the targets we are interested in. However, from a model development perspective, a clear outcome, such as positive cultures, can more clearly evaluate the model compared to sepsis, which has a definition, but is more difficult to define clinically.

Our ultimate goal is that our machine learning model identifies the appropriate antimicrobials for patients being treated with empiric therapy for various conditions, including sepsis, which we aim to incorporate into our model in future studies.

2. The authors present data in Table 4 suggesting possible benefits of using the Deep Learning Model compared with clinical judgment. I was not clear how "clinical judgment" was determined - was this based on the use of certain antibiotics during patients' treatment? If so, how did the authors check if the antibiotic was given for MRSA or something else?

Thank you for your comment. Yes, this is a weakness of our study. Since we wanted to be as accurate as possible, we decided to use two antimicrobial categories, MRSA-specific and any-MRSA antimicrobials. (Addressed in Possible Clinical Impact of Our Model subsection under Methods).

MRSA specific antibiotics are frequently (not exclusively) used to empirically target or treat MRSA infections. Vancomycin is usually used to specifically treat MRSA. The other antimicrobials, like daptomycin or linezolid, can be used to target vancomycin-resistant enterococci. We only found 6 cases in MHHS and 32 cases in MIMIC-IV test datasets, which we used to evaluate the possible clinical impact. To treat other gram-positive cocci such as Streptococci, other antimicrobials such as cephalosporins or penicillin class are sufficient. Further, we decided to look into MRSA bacteremia which is almost always true infection when MRSA is identified from blood cultures. Even in this patient population, we found similar results as other types of patient population.

To clarify this point, we added the number of VRE cases in test datasets in the manuscript. We also added sentences in the weakness.

3. Assuming the authors can provide reassurance regarding point 2, if they have adequate data on clinical judgment, could the authors not compare the AUROC curve for Deep Learning Model vs. clinical judgment? Indeed, would this not be a more appropriate comparison (clinically) than with models generated using different methods?

Since clinical judgement does not provide the numerical estimation of clinician's judgements only binary whether they started antimicrobials or not, we cannot obtain AUC curve for physician's judgement. Instead, we identified overall sensitivity and specificity of clinicians' decision if they started MRSA specific antibiotics and any MRSA antibiotics. We plotted the results in our AUC curves. (Supplement Figure 5)

Indeed, would this not be a more appropriate comparison (clinically) than with models generated using different methods?

Thank you for your comment. Yes, our final goal is to compare our model versus clinician's judgement. To achieve this, we believe a prospective study will be more appropriate. Toward this goal, in this project, we aimed to show that by integrating time series data, our deep learning model is more similar to the physician's decision-making process compared to other types of models. Furthermore, we wanted to see how our model can improve the physician's decision-making in retrospective data. In future studies, we aim to prospectively compare our deep learning-based model against physicians' decisions.

Reviewer #3 (Remarks on code availability):

I have no expertise in the writing or interpretation of code.